# Recent Advances in Signaling Pathways Comprehension as Carcinogenesis Triggers in Basal Cell Carcinoma

**DOI:** 10.3390/jcm9093010

**Published:** 2020-09-18

**Authors:** Mircea Tampa, Simona Roxana Georgescu, Cristina Iulia Mitran, Madalina Irina Mitran, Clara Matei, Cristian Scheau, Carolina Constantin, Monica Neagu

**Affiliations:** 1Department of Dermatology, “Carol Davila” University of Medicine and Pharmacy, 050474 Bucharest, Romania; tampa_mircea@yahoo.com (M.T.); matei_clara@yahoo.com (C.M.); 2Department of Dermatology, “Victor Babes” Clinical Hospital for Infectious Diseases, 030303 Bucharest, Romania; 3Department of Microbiology, “Carol Davila” University of Medicine and Pharmacy, 050474 Bucharest, Romania; cristina.mitran@drd.umfcd.ro (C.I.M.); irina.mitran@drd.umfcd.ro (M.I.M.); 4Department of Physiology, “Carol Davila” University of Medicine and Pharmacy, 050474 Bucharest, Romania; cristian.scheau@umfcd.ro; 5Immunology Department, “Victor Babes” National Institute of Pathology, 050096 Bucharest, Romania; caroconstantin@gmail.com (C.C.); neagu.monica@gmail.com (M.N.); 6Colentina Clinical Hospital, 020125 Bucharest, Romania; 7Faculty of Biology, University of Bucharest, 76201 Bucharest, Romania

**Keywords:** basal cell carcinoma, Hedgehog pathway, signaling pathways, carcinogenesis

## Abstract

Basal cell carcinoma (BCC) is the most common malignant skin tumor. BCC displays a different behavior compared with other neoplasms, has a slow evolution, and metastasizes very rarely, but sometimes it causes an important local destruction. Chronic ultraviolet exposure along with genetic factors are the most important risk factors involved in BCC development. Mutations in the *PTCH1* gene are associated with Gorlin syndrome, an autosomal dominant disorder characterized by the occurrence of multiple BCCs, but are also the most frequent mutations observed in sporadic BCCs. PTCH1 encodes for PTCH1 protein, the most important negative regulator of the Hedgehog (Hh) pathway. There are numerous studies confirming Hh pathway involvement in BCC pathogenesis. Although Hh pathway has been intensively investigated, it remains incompletely elucidated. Recent studies on BCC tumorigenesis have shown that in addition to Hh pathway, there are other signaling pathways involved in BCC development. In this review, we present recent advances in BCC carcinogenesis.

## 1. Introduction

Basal cell carcinoma (BCC) is the most prevalent form of skin cancer, developing on sun exposed areas, especially in the fourth decade of life. BCC is a slow-growing, locally invasive tumor, with a low capacity of metastatic spread [1,2]. It is commonly recognized that only 0.0028–0.55% of BCCs will metastasize [3]. Exposure to ultraviolet (UV) light is a key factor in its pathogenesis [4]. Therefore, most cases of BCC are diagnosed in individuals with fair skin phototypes that carry out activities involving intense, intermittent or continuous exposure to UV [5]. In addition, the exposure to ionizing radiation, arsenic or coal tar derivatives increases the risk of developing a BCC. The incidence of BCC is higher compared to the general population in two particular scenarios, immunosuppressed patients and patients with certain genodermatoses such as Gorlin syndrome [6,7]. In this review, we focused on recent advances related to the signaling pathways involved in BCC carcinogenesis.

## 2. The Genetic Basis of Basal Cell Carcinoma Initiation and Therapy Resistance

### 2.1. Genes Involved in Nevoid Basal Cell Carcinoma Syndrome

Nevoid basal cell carcinoma syndrome (NBCCS) is an autosomal dominant disorder characterized by mutations in the *patched* (*PTCH*)*1* gene, *PTCH2* gene and *suppressor of the fused* (*SUFU*) gene, which are negative regulators of the hedgehog (Hh) pathway [8]. *PTCH1*, located on chromosome 9q22.3, encodes the homologous transmembrane protein PTCH1 that acts as a receptor for the Hh pathway [9]. *PTCH2* is located on chromosome 1p34 and encodes for PTCH2 and *SUFU* is located on chromosome 10q24.32 and encodes for the suppressor of a fused homologous protein, SUFU [8]. The prevalence of NBCCS ranges between 1/57.000 and 1/256.000. It is a multisystemic disease characterized by the development of multiple BCCs, jaw keratocysts, palmar and plantar pits, abnormalities of the bones and eyes, cardiac dysfunction, calcification of the falx cerebri, etc. In about 5% of cases patients can associate intellectual deficiency [10]. NBCCS is also known as Gorlin Goltz syndrome, after the name of those who described it as a distinct entity in 1960. It occurs most commonly in Caucasian adults aged 17–35 years, with no sex predominance [11].

### 2.2. Genes Involved in Sporadic BCC

Sporadic BCC is also related to genetic alterations in components of the Hh pathway. Mutations in the *PTCH1* gene were observed in 30–60% of cases, in the *smoothened* (*SMO*) gene in 10–20%, and to a lesser extent in the *SUFU* gene. Alterations involving glioma-associated oncogenes (*GLI*) are rare [12]. Mutations in *TP53* gene were observed in a high number of cases, over 50% of BCCs. *TP53* encodes for the p53 protein, one of the most important regulators of the cell cycle. Mutations in *TP53* seem to be involved in the initiation of the malignant process but also in tumor progression [13].

Given that BCC has a great diversity in terms of clinical appearance, histopathological forms, evolution and response to treatment, Bonilla et al. considered that there are many other genes involved in its pathogenesis. Thus, they identified mutations in *MYCN*, *PTPN14,* and *LATS1*. *MYCN* alterations were observed in 30% of the studied BCC samples, most of them being identified in the Myc box 1 (MB1) region. Mutations in *PTPN14* were observed in 23% of cases, and in *LATS1* in 16% of cases [14].

Moreover, alterations in pigmentary genes were detected in BCC patients [13]. Genetic studies have revealed several BCC susceptibility regions such as 1p36, 1q42, 5p13.3, 5p15, and 12q11-13. A recent study has found new susceptibility regions on chromosome 5, 5q11.2-14.3, 5q22.1-23.3, and 5q31-35.3. These findings may underlie the development of new diagnostic tools and therapeutic approaches in BCC management [15].

A large number of mutations have been revealed so far in BCC cells, therefore Jayaraman et al. hypothesized that this variety of mutations leads to the activation of the host’s defense system, which may explain why BCC evolves slowly and metastasizes very rarely [16]. In line with this, the study by Dai et al. performed on 19 BCC samples has revealed the overexpression of 222 genes and the downregulation of 91 genes. Upregulated genes were involved in cell cycle regulation and mitosis, while downregulated genes were involved in cell differentiation and unsaturated fatty acid metabolism. The increased expression of cyclin-dependent kinase (CDK)-1, a regulator of the cell cycle, has been observed, and may represent a novel target for new therapies in BCC [17].

### 2.3. Genes Linked to Therapy Resistance

There are several attempts to target Hh pathway, some of them already approved in BCC, some of them in the preclinical phase (Table 1). Gene mutations plays an important role in the response to drug therapy. About 20% of patients with BCC treated with vismodegib, a SMO inhibitor, undergo treatment failure within one year of treatment. The main mechanism involved in resistance development is the overexpression of several components of the Hh pathway [18]. Mutations were observed in both the vicinity and distally of drug binding situs of SMO. SMO mutations that occur in the vicinity of the drug binding domain such as D473, H231, W281, Q477, V321, I408, and C469, have been detected only in resistant BCCs which suggests the role of the drug therapy in acquiring these mutations. Mutations distal to the drug binding domain such as T241M, A459V, L412F, S533N, and W535L were found in both untreated BCCs and resistant tumors revealing their inherent role [19].

Vismodegib resistance in BCC was also linked to mutations in *TP53* [18]. Mutations in *SUFU* were linked to resistance to vismodegib in a small number of cases [3]. However, tumor resistance was identified in patients without an identifiable mutation [30]. Secondary resistance to vismodegib was first described in a patient diagnosed with medulloblastoma [31].

The mechanisms involved in BCC resistance are not only related to mutations in the canonical Hh pathway, therefore Whitson et al. have shown in a mouse model that the activation of non-canonical Hh pathway by MKL1/SRF is related to the resistance to SMO inhibitors in some BCCs. Thus, they have highlighted the role of MKL1 inhibitors in the treatment of BCC in combination with SMO inhibitors, MKL1 inhibitors could exhibit a synergistic effect [32].

## 3. Hedgehog Pathway—From Discovery to New Concepts

The Hh pathway plays an essential role in human embryogenesis, being involved in cell differentiation, cell growth, and morphogenesis. [33]. Under normal conditions, the hair follicle and the skin are the only two regions where the Hh signaling displays post-embryonic activity. Hh signaling is also active in stem cells and in tissues undergoing regeneration, having an important role in wound healing [13]. The ectopic activation of the Hh pathway contributes to tumorigenesis, metastasis and resistance to therapy [34]. The first link between BCC and the Hh pathway was revealed in the context of the discovery of loss-of-function mutations in *PTCH1* gene in patients with Gorlin syndrome [35].

Recent research has revealed that Hh signaling can be activated through different pathways [36]. Thus, Hh signaling was classified as canonical and non-canonical [37]. The canonical Hh pathway involves a GLI-mediated transcription. When the activation of Hh pathway occurs independently of GLI-mediated transcription it is categorized as non-canonical Hh pathway [38]. The aberrant stimulation of the Hh pathway as a result of mutations in *PTCH1* and *SMO* is involved in the development of BCC [39]. The binding of one of the Hh ligands to PTCH1, a 12-pass transmembrane receptor protein that prevents the activation of Hh pathway, is the first step required for the activation of the canonical Hh pathway. In vertebrates, three ligands were described, including Sonic hedgehog (Shh), Indian hedgehog (Ihh), and Desert hedgehog (Dhh), of which Shh is the strongest pathway activator [37]. Hh bindings proteins, such as Hh interacting proteins, sequester Hh ligands and in this manner control the amount of Shh that binds to PTCH1 [40]. The Hh ligands bind to PTCH1 and remove it from the primary cilium resulting in the stimulation of SMO, a 7-pass transmembrane protein, and its translocation to the primary cilium. The accumulation of SMO triggers a cascade of events that promote the transcriptional activation of GLI, resulting in cell proliferation (Figure 1) [39,41].

Recent studies have shown that there are two different categories of non-canonical Hh signaling, type 1 acting via PTCH1, in a SMO independent manner and type 2 acting via SMO, independently of GLI regulation. The role of non-canonical Hh signaling in skin cancers is not fully elucidated [38].

Non-canonical pathways involved in BCC tumorigenesis include K-Ras, transforming growth factor-β (TGF-β), PI3K/Protein kinase B (AKT)/mammalian target of rapamycin (mTOR), protein kinase C, and the serum-response factor-megakaryoblastic leukemia-1 pathway [34,37].

In the absence of Hh ligands, PTCH1 is located on the primary cilium and does not allow SMO migration and insertion into the primary cilium. The GLI transcription factors are phosphorylated by protein kinases and undergo proteolytic cleavage resulting in repressor molecules that will suppress the activation of the Hh pathway [42,43]. In other words, Hh ligands are the initiators of Hh pathway, PTCH1 operates as a negative regulatory receptor, and SMO as a positive regulatory receptor. In the absence of Hh ligands, PTCH1 binds to SMO preventing the pathway activation [13,44]. GLI transcription factors are blocked into the cytoplasm by various proteins acting as mediators. The most important proteins involved are protein kinase A (PKA) and SUFU. The proteolytic cleavage of GLI transcription factors generates the repressor forms, GLI2R and GLI3R. The mechanism by which PTCH1 suppresses SMO function in the absence of ligands is not fully known. There have been postulated several theories. It seems that PTCH1 does not allow SMO activation by blocking SMO agonists, such as primary cilium oxysterols. Another theory claims that PTCH1 increases the influx of SMO antagonists into the primary cilium [45].

## 4. The Role of Inflammation and Immune Response in BCC Pathogenesis

A substantial body of evidence indicates that BCC is an immunogenic tumor. This is supported by the increased incidence of BCC among immunosuppressed subjects and by the presence of numerous immune cells that infiltrate the tumor and peritumoral area [46].

### 4.1. Tumor Microenvironment in BCC

In BCC samples, it has been observed a high number of immature CD11c + myeloid dendritic cells (DCs) which suggests that tumor microenvironment has an immunosuppressive effect, knowing that immature DCs induce downregulation of T cells [47]. Regulatory T cells (Tregs) have the ability to prevent the maturation of DCs [48] and it should be highlighted that an increased number of FOXP3 + Treg have been identified in the tumor and peritumoral area, and not identified in normal skin. However, the role of Tregs in BCC is not fully understood. [49]. The cell composition of the inflammatory infiltrate may be influenced by various factors, UV exposure or treatment. It has been shown that, after exposure to UV light, Langerhans cells stimulate the function of the Th2 subset of CD4+ T cells [50]. After immunocryosurgery, a significant increase in the CD3+/Foxp3+ ratio was observed, which denotes the induction of an antitumor response [51]. Tumor-associated macrophages (TAMs) also exert an immunosuppressive effect in BCC, their presence being associated with increased invasion capacity and high microvessel density [47].

The increased expression of Th2 cytokines is an additional factor in the generation of an immunosuppressive tumor microenvironment. High levels of IL-4 and IL-10, type 2 cytokines, have been identified in BCC [48]. High levels of IL-10 correlate with a decreased expression of MHC-I and other molecules such as ICAM-1, CD40, CD80, and HLA-ABC. In BCC the low number of CD8 cells and decreased expression of MHC-I allow the tumor escape from immune surveillance. Treatment with Hh inhibitors is associated with an increase in the number of CD8+ and CD4+ T cells [52]. A recent study suggests a potential role of IL-23/Th17-related cytokines in BCC. The role of IL-23 in carcinogenesis is not fully elucidated, increased levels of IL-23 being associated with both tumor growth and apoptosis. In contrast, in regressing BCC, a Th1 immune response has been revealed [53].

### 4.2. The Link between Inflammation and Hh Signaling

Recent research highlights the link between Hh signaling and immune cells. Data obtained from a study conducted on murine BCC cells, have revealed that Hh signaling induces the migration of myeloid-derived suppressor cells (MDSC) and M2 polarization of macrophages, resulting in an immunosuppressive tumor microenvironment. Moreover, keratinocytes presenting SMO oncogene release TGF-β with an inhibitory action on effector T cells. Another immune-related mechanism in which Hh signaling is involved is the decrease in MHC-I molecule expression on the cell membrane of malignant cells, a phenomenon that hinders immune system recognition [54]. A recent study has shown that the inhibition of the Hh pathway in BCC patients treated with vismodegib or sonidegib (SMO inhibitors) resulted in an increased MHC-I expression in tumor cells associated with a high number of CD4 and CD8 T cells infiltrating the tumors [55]. Given the immunosuppressive effect of Hh signaling, it was found useful to associate Hh inhibitors with immune checkpoint inhibitors. Thus, patients with BCC who received nivolumab or pembrolizumab obtained encouraging results [54]. Hh signaling seems to reduce TCR signaling in mature T cells, and the inhibition of Hh signaling promotes T cell activation and proliferation and hence induces an anti-tumoral effect [55]. The relationship between inflammation and carcinogenesis has been intensively studied in the last decade [50,56]. Chronic inflammation is a key factor in cell malignant transformation [57]. Pro-inflammatory cytokines are important players in the initiation and perpetuation of the inflammatory process [58]. Interleukin 6 (IL-6) is the pro-inflammatory cytokine prototype [59,60]. A recent study has shown that IL-6 stimulates tumorigenesis by synergistically acting with Hh pathway. Hh—IL-6 signaling tandem is based on the activation of the signal transducer and activator of transcription 3 (STAT3) via IL-6/Jak2 pathway. IL-6 and Hh pathway interact at the level of cis-regulatory regions following the cooperation of GLI and STAT3. Regarding the activation of IL-6 signaling, three mechanisms have been suggested, including its activation by Hh pathway, its activation under the influence of the tumor microenvironment or via sIL6R-mediated trans-signaling [61].

## 5. Crosstalk between Hh Signaling Pathway and Other Signaling Pathways in BCC

BCC carcinogenesis is orchestrated by various signaling pathways that cooperate and form a complex network [62] (Figure 2).

### 5.1. Wnt/β-Catenin Pathway

Wnt proteins are a complex of 19 lipidated and glycosylated proteins, which govern the activity of the canonical, β-catenin-dependent, and non-canonical, β-catenin-independent, Wnt pathways. In the non-canonical pathway, β-catenin does not undergo activation [63]. The Wnt/β-catenin pathway mediates numerous processes such as cell proliferation, migration and invasion and is involved in the development of several cancers. β-catenin is a member of a multi-molecular complex consisting of axin, adenomatous polyposis coli (APC) and glycogen synthase kinase β (GSK3β). In the absence of Wnt signaling, β-catenin is phosphorylated by GSK3β, ubiquitinated and subsequently it is degraded into the proteasome. When a Wnt ligand binds to the Frizzled receptors and the low-density lipoprotein receptor related protein (LRP), GSK3β is inactivated, β-catenin escapes from the complex and is translocated to the nucleus where Wnt target genes are upregulated [64,65]. Mutations in the Wnt pathway can lead to its activation independently of ligands and subsequently the malignant process is initiated. Under normal conditions, the Wnt pathway is inactive [65].

The Hh pathway downregulates the Wnt pathway through secreted frizzled-related protein 1 (SFRP1), and the Wnt pathway modulates the activity of the Hh pathway through GLI3. A disruption in this antagonism may be involved in tumorigenesis [66]. In BCC, the activation of the Hh pathway can induce aberrant activation of the Wnt pathway by the GLI transcription factors. The crosstalk between the two pathways is mediated by several molecules. Wnt2b, Wnt4, and Wnt7b are activated by GLI1 and subsequently the Wnt/β-catenin pathway is stimulated. In addition, β-catenin can increase the expression of the coding region determinant-binding protein (CRD-BP) and thus promotes the stabilization of GLI mRNAs [65]. Noubissi et al. showed that in BCC the expression of CRD-BP is increased and there is a positive correlation between CRD-BP level and the activation of Wnt and Hh signaling pathways. A decreased CRD-BP expression is linked to a low proliferation rate of BCC cells [67]. Alternatively, GLI1 simulates Wnt proteins and Snail to promote the translocation of β-catenin from the cell’s membrane to its nucleus; in the cell membrane, β-catenin forms a complex with E-cadherin and Snail acts as a suppressor of E-cadherin (Figure 2) [68]. In about 30% of BCC samples, it has been found an accumulation of β-catenin in the nucleus, an accumulation that is associated with a higher proliferation rate [69]. GSK3β is not only a member of the Wnt pathway, but it has also been demonstrated that GSK3β participates in the Hh pathway as well, being involved in the activation of Snail by GLI1 [68]. In the skin, the Wnt pathway may also interact with the Notch and vitamin D pathways [63].

However, Carmo et al. have found a downregulation of Wnt3 and Wnt16 in 58 nodular BCC samples when compared to healthy tissue. Wnt3 can activate both canonical and non-canonical pathways and is involved in cell proliferation and malignant transformation. Wnt3 overexpression has been commonly identified in aggressive tumors. Carmo et al. pointed out that BCC is not an aggressive tumor, therefore there may be different gene expression profiles in such tumors [70].

Brinkhuizen et al. have shown that promoter hypermethylation of the components of the Hh and Wnt pathways is involved in carcinogenesis in BCC, a finding that may underlie the development of new therapies [12]. Interestingly, it appears that in some cases of BCC treated with Hh inhibitors, the Wnt pathway could play a role in relapse by modulating the transcriptional profile of the residual cells [69].

### 5.2. PI3K/AKT/mTOR Pathway

The PI3K family comprises enzymes with multiple subunits that act jointly to induce the conversion of phosphatidylinositol diphosphate (PIP2) to phosphatidylinositol triphosphate (PIP3). PIP3 via phosphoinositide-dependent kinase-1 (PDK-1), promotes the phosphorylation of AKT, a serine/threonine kinase, and its conversion to the active form. AKT can act on many targets, one of the main targets being mTOR. Other important targets are cyclic AMP–responsive element binding protein (CREB), and procaspase 9, p21, p27 families [71]. The PI3K/AKT signaling pathway is involved in mTOR phosphorylation. The downstream effector of PI3K is mTOR, which also acts as an upstream regulator [72].

The PI3K/AKT/mTOR pathway plays an important role in the normal growth and development of the human body. Mutations in the components of this pathway can lead to alterations that modify mTOR signaling, therefore the aberrant mTOR signaling pathway has been identified in various disorders [73,74]. mTOR mediates cell growth and alterations in mTOR pathway have been related to the development of several other neoplasms [72]. Certain growth factors and oncogenic proteins act as activators of the PI3K/AKT/mTOR pathway. The stimulation of PI3K/AKT/mTOR signaling promotes the phosphorylation and activation of several protein kinases, which are involved in carcinogenesis [62].

mTOR is a serine/threonine kinase that pertains to the PI3K-related protein kinase family; its C-terminus exhibits a great structural similarity to the catalytic domain of PI3K. mTOR includes two protein complexes, mTOR complex 1 (mTORC1), and mTOR complex 2 (mTORC2), that have different functions. mTORC1 is upregulated by the PI3K/AKT signaling and downregulated by the TSC1/TSC2 complex. The downstream targets of mTORC1 are S6K1 and 4EBP1, which control mRNA translation. mTORC2 is upregulated by growth factors, activates PKC-α and AKT and regulates the function of the small GTPases (Rhoa, Rac1 and Cdc42), involved in cell survival and modulation of the actin cytoskeleton [73,75].

Kim et al. highlighted that in BCC there is a crosstalk between Hh and PI3K/AKT/mTOR pathways. SOX9, a protein whose expression is mediated by GLI, stimulates mTOR transcriptional activity. Moreover, depletion of SOX9 is associated with a decreased mTOR expression and consequently a decreased BCC cell proliferation [76]. PI3K induces PDK1 activation which in turn will activate S6K1. S6K1 can phosphorylate GLI1, thus GLI1-SUFU interaction is blocked, GLI1 is translocated to the nucleus inducing GLI-dependent transcription (Figure 2) [42].

The inhibition of mTOR is associated with the activation of another important cellular process, autophagy [43]. The latest studies point out that the role of autophagy in tumorigenesis should be studied more deeply as autophagy is interconnected with Hh signaling. Autophagy is an important process responsible for the elimination of damaged cells, being involved in tumor initiation and progression. It seems that Hh signaling has both a stimulatory and inhibitory effect on autophagy, but most studies have revealed its inhibitory role [34].

Everolimus, an immunosuppressive agent, acting on mTOR has shown encouraging results in BCC therapy [77]. The use of inhibitors of PI3K/AKT/mTOR pathway in combination with SMO inhibitors may enhance the effect of SMO inhibitors leading to a better response in BCC [78].

### 5.3. Hippo-YAP Pathway

The Hippo-YAP pathway mediates important cell processes such as cell differentiation, proliferation and apoptosis and through its downstream effectors, YAP and TAZ, is responsible for skin barrier function. In the damaged skin areas YAP and TAZ activate the stem cells involved in tissue regeneration [79]. YAP and TAZ play an essential role in embryonic development. At the same time, YAP and TAZ may contribute to carcinogenesis through the activation of target genes that promote cell proliferation, epithelial-to-mesenchymal transition (EMT) and metastasis [80].

Recent research has revealed that in BCC, the overexpression of the Hippo-YAP pathway participates in the process of tumorigenesis [79]. YAP and TAZ are two molecules that shuttle between the nucleus and the cytoplasm. In the nucleus, YAP and TAZ stimulate the expression of proliferative and antiapoptotic genes following the interaction with transcriptional factors of the TEA domain family members (TEAD). It has been observed that aberrant activation of the nuclear form of YAP is associated with basal cell proliferation and decreased markers of cell differentiation. However, the mechanism by which YAP initiates carcinogenesis in BCC is still unknown. Mutations in some genes, that control YAP and TAZ phosphorylation—*LATS1* and *LATS2* and their translocation from the nucleus to the cytoplasm—*PTN14*, are involved in the aberrant activation of the Hippo signaling [81]. The role of the Hippo pathway in BCC carcinogenesis is supported by the study conducted by Bonilla et al. which analyzed 293 BCC samples and showed that YAP target genes are overexpressed [14]. The study performed by Maglic et al. has found that Hippo signaling induces BCC carcinogenesis via the c-JUN/AP1 axis [82].

In a mouse model, Akladios et al. have revealed that positive regulatory interactions between YAP and Hh signaling are involved in BCC development. They showed that epidermal YAP activity induces the accumulation of GLI2 into nucleus in YAP2-5SA-ΔC mice [83].

### 5.4. EGFR Pathway

The epidermal growth factor receptor (EGFR) belongs to the ErbB family of tyrosine kinase receptors and stimulates the growth of cells previously activated by an EGFR ligand [84]. The specific ligands of EGFR are the epidermal growth factor, amphiregulin, TGF or heparin growth factor [85]. The binding of soluble ligands to the ectodomain of the receptor leads to homo and heterodimerization with other members of the receptor family. Receptor dimerization is a key step for the activation of its intracellular tyrosine kinase domain. Phosphotyrosine residues activate signaling pathways including Ras/MAPK, PLCγ1/PKC, PI3K/AKT, and STAT pathways [86]. EGFR overexpression is found in various tumors and represents an important promoter for the activation of different signaling pathways, leading to cell proliferation, invasion and metastasis [86]. EGFR is involved in some cases of SCC; 7% of head and neck SCC (HNSCC) display EGFR mutations [87].

Recent studies attribute a role to the EGFR pathway in BCC. Avci et al. detected a high EGFR expression in BCC samples, identifying a 4.17-fold increased expression in tumoral tissue compared to healthy tissue. In addition, the EGFR expression was 6.66 times higher in recurrent BCC compared with non-recurrent BCC. Analyzing the histopathological type, they concluded that EGFR can be considered a negative prognostic marker for infiltrative BCC with important consequences in terms of resection margins. The results were not statistically significant in the case of nodular and superficial BCC [88]. Similarly, another study found an increased EGFR expression in the analyzed BCC samples. The highest expression of EGFR was identified in the adenoid and morpheiform types and the lowest in the nodular type, suggesting that EGFR plays a role in the histological differentiation of BCC [85].

In vitro studies have shown that the interaction between Hh and EGFR pathways modulates the Hh target genes. The cooperation between EGFR and Hh signaling promotes the activation of RAS/MEK/ERK and JUN/AP-1 signaling (Figure 1). EGFR/Hh signaling is involved in the up-regulation of several genes required for BCC development including *SOX2, SOX9, JUN, CXCR4,* and *FGF19* [89]. Moreover, EGFR by activating ERK1/2 suppresses GLI2 proteolytic degradation in keratinocytes [43].

The study performed by Schnidar et al. emphasized the usefulness of a therapy based on the combined inhibition of the Hh and EGFR pathways. It has been observed that BCC cells express certain EGFR ligands, indicating the autocrine stimulation of this pathway [90]. Therapy with cetuximab, a monoclonal antibody that inhibits EGFR, has revealed promising results in keratinocyte carcinomas (BCC and SCC) [91].

### 5.5. Vitamin D Pathway

The action of vitamin D in cancer seems to be dual, with both pro- and anti-carcinogenic effects. The activation of vitamin D receptor (VDR) in the skin induces an antiproliferative effect by stimulating or inhibiting different pathways. [92,93]. In the skin, vitamin D inhibits the Hh signaling pathway as a protective mechanism against the harmful effects of UVB radiation. It has been observed that, in *Vdr*-null mice, the Hh pathway is overexpressed in the epidermis and hair follicle. Lack of VDR in keratinocytes interferes with cell differentiation, tissue repair, and increases the risk of developing a malignant process [94]. Teichert et al. have shown that vitamin D may directly inhibit the Hh pathway in a VDR-dependent manner. However, vitamin D might inhibit Hh pathway independently of VDR [95].

Recent studies have shown that vitamin D suppresses the Hh pathway by inhibiting SMO function. The mechanism is not fully understood but it has been shown that vitamin D acts upstream of PTCH and downstream of GLI. Another argument that vitamin D represses the Hh pathway by inhibiting SMO function is that there is no inhibition of the Hh pathway in the case of *SMO*-null cells [96]. Thus, Uhman et al. have shown that the application of calcitriol, the active form of vitamin D3, on the skin represses the development of BCC in *Ptch* mutant mice [97]. Calcitriol activates the VDR signaling pathway resulting in an antiproliferative effect and mediates cell differentiation by increasing the expression of markers such as involucrin, loricrin, and filaggrin. Moreover, calcitriol can mediate skin apoptosis [98]. Calcitriol is secreted by fibroblasts and released under the action of PTCH. It was pointed out that in *PTCH*-null cells, the synthesis of calcitriol occurs but the compound cannot be released [96].

However, the study by Brinkhuizen et al. did not reveal the efficacy of calcitriol in the treatment of superficial BCC. They have also tried a combination between diclofenac and calcitriol to reveal a synergistic effect, but there were no results. However, the use of diclofenac 3% gel in hyaluronic acid in BCC promotes apoptosis and inhibits cell proliferation [99].

### 5.6. P53 Pathway

There are several studies providing data on the role of *p53* in BCC pathogenesis. It has been shown that *p53* is overexpressed in BCC samples and suggested that *p53* mutations following chronic UV exposure might be an important factor in BCC development [100]. *P53* is a well-known tumor suppressor gene and has important implications for cancer prevention; therefore, mutations in the *p53* gene have been identified in various neoplasms. P53 protein may undergo inactivation by interacting with various proteins such as MDM-2, MDMx, and FAK [101]. It acts as a transcription factor by binding to certain sequences in the DNA structure leading to the activation or suppression of target genes. Thus, p53 controls pathways involved in cell division and DNA repair [102].

The role of *p53* mutations in BCC pathogenesis is not clear. In this regard, Oh et al. conducted the first study that showed an increased expression of p53, ΔNp63, TAp73, and γ-H2AX associated with the downregulation of MDM-2 [103]. The *p63* gene has two isoforms, TAp63 acts as a suppressor, whereas ΔNp63 acts as an oncogene. Similarly, the *p73* gene has two isoforms, TAp63, with tumor suppressor effect and ΔNp73 with an oncogenic role. Exposure to UVB leads to the production of γ-H2AX, which can be regarded as a marker of UVB-related DNA damage. MDM-2 is a negative modulator of p53 [103,104]. The study by Wang et al. tried to answer the question regarding the mechanism by which p53 is activated in BCC. The study has shown that aberrant Hh signaling activates p53 via Arf. In addition, the study has revealed that loss of p53 results in tumor development and progression. In contrast, loss of Arf is not associated with the initiation of the malignant process but is involved in tumor progression. On the other hand, an increased Arf expression in tumor keratinocytes contributes to the suppression of BCC development. The stress induced by oncogenes results in Arf activation which induces an increased p53 expression [105]. Arf/p53 pathway is involved in the elimination of altered cells [106].

The study by Li et al. has revealed that one of the mechanisms by which Hh pathway is involved in BCC tumorigenesis is the evasion of p53 activity. Moreover, Hh signaling contributes to p53 degradation in mouse embryonic fibroblasts [107].

Alterations of p53 function have implications for the treatment of BCC. A recent study on a BCC cell line has found that imiquimod promotes reactive oxygen species production, which will stimulate ATM and ATR signaling pathways contributing to cell apoptosis mediated by p53. In addition, cell lines that displayed mutations of p53 were more resistant to imiquimod-induced apoptosis [108].

### 5.7. Notch Pathway

The Notch signaling pathway could be involved in BCC tumorigenesis. Notch receptors are a group of four transmembrane proteins (Notch1-4) that are able to interact with different ligands. The most popular ligands are jagged 1 and 2 and delta 1, 3, and 4 [109]. The binding of the specific ligands promotes the activation of Notch signaling. This interaction produces the intramembrane cleavage of Notch receptor resulting in the release of the intracellular fragment of Notch (NICD), which translocates to the nucleus and activates the expression of Notch target genes [110].

Recent studies have shown that the interaction between Hh and Notch signaling pathways is involved in carcinogenesis and resistance to chemotherapy [111]. It has been found that Notch1 acts as an inhibitor of a malignant process. A study on animal models with Notch1-deficient skin has revealed a spontaneous development of BCC after a certain period. Moreover, in these cases, an increased activation of the Hh pathway was observed [109].

Shi et al. pointed out that there is a low expression of the Notch signaling pathway in BCC. After stimulating Notch signaling with Notch signaling peptide jagged 1, BCC cells undergo apoptosis. Interestingly, an increased activity of the Notch pathway was observed in the hair follicle, the origin of BCC. Thus, it was hypothesized that studying the Notch pathway in BCC may allow the introduction of new therapies [112]. Eberl et al. analyzed Notch expression in BCC cells and observed that the inner cells that display increased Notch activation after vismodegib treatment die, while those in the periphery that do not express Notch survive and lead to tumor recurrence. Thus, it seems that Notch modulation plays an important role in the pathogenesis and treatment of BCC [111].

Moreover, there is an important crosstalk between Notch and Wnt pathway. Wnt stimulates the expression of the Notch ligand Jagged, in turn Notch exerts an inhibitory effect on Wnt expression. In addition, mastermind-like protein 1 (Maml1), a coactivator of the Notch signaling, may act as a regulator of β-catenin transcription (Figure 2) [113].

## 6. Conclusions

The pathogenesis of BCC is very complex. *PTCH1* mutations play a crucial role in activating the Hh pathway; however, additional mutations that promote BCC carcinogenesis have been identified. Recent studies have shown that there is a significant cross-talk between Hh signaling pathway and other signaling pathways, including Wnt, Notch, EGFR, p53, PI3K/mTOR, and vitamin D. A further argument for the involvement of other pathways in the development of BCC could be the tumor resistance to Hh inhibitors. The knowledge and characterization of the BCC signaling pathways and the interactions between them could underlie the development of new therapies in BCC.

## Figures and Tables

**Figure 1 jcm-09-03010-f001:**
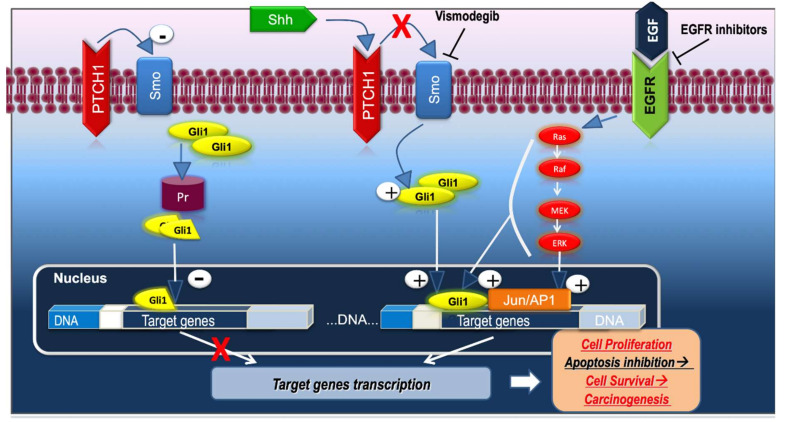
Hh pathway (inactive state and active state) and the crosstalk between Hh and EGFR pathways. Shh—Sonic hedgehog; PTCH1—protein patched homolog 1; SMO—smoothened protein; GLI—glioma-associated oncogenes; Pr—proteasome; EGFR—epidermal growth factor receptor; Ras—rat sarcoma virus; Raf—rapidly accelerated fibrosarcoma; MEK—mitogen-activated protein kinase kinase; ERK—extracellular signal-regulated kinase.

**Figure 2 jcm-09-03010-f002:**
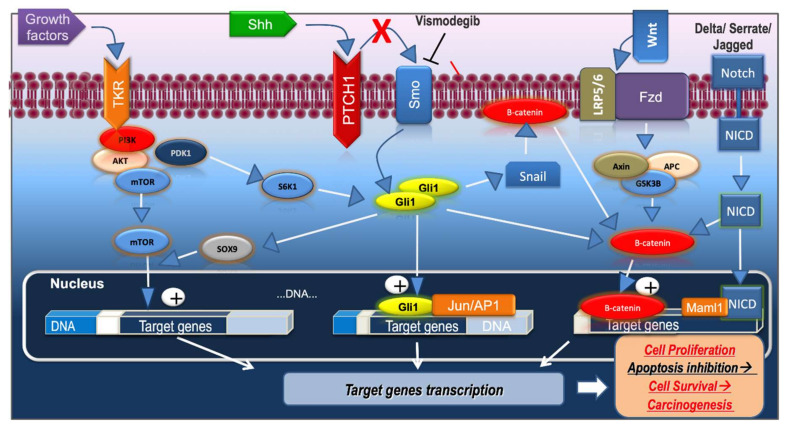
Crosstalk between Hh signaling pathway and PI3K/AKT/mTOR pathway, Wnt/β-catenin pathway, Notch pathway. Shh—Sonic hedgehog; PTCH1—protein patched homolog 1; SMO—smoothened protein; GLI—glioma-associated oncogenes; TKR—tyrosine kinase receptor; PDK-1—phosphoinositide-dependent kinase-1; PI3K—phosphoinositide 3-kinase; AKT—protein kinase B (PKB); mTOR—mammalian target of rapamycin; S6K1— S6 kinase 1; FZD—Frizzled receptors; LRP—low-density lipoprotein receptor related protein; APC—adenomatous polyposis coli; GSK3β—glycogen synthase kinase β; NICD—the intracellular fragment of Notch; Maml1—mastermind-like protein 1.

**Table 1 jcm-09-03010-t001:** Hedgehog inhibitors in basal cell carcinoma.

Target	Therapy Molecule	Reference
SMO	Vismodegib *	Sekulic et al. [20]
SMO	Sonidegib **	Danial et al. [21]
SMO	Itraconazole	Kim et al. [22]
SMO	BMS-833923	Siu et al. [23]
SMO	Taladegib	Bendell et al. [24]
SMO	Patidegib	Jimeno et al. [25]
SMO	NVP-LQ506	Peukert et al. [26]
SHH	Robotnikinin	Hassounah et al. [27]
GLI	GANT-58 and GANT-61	Lauth et al. [28]
GLI	Arsenic trioxide	Ally et al. [29]

* Approved by the FDA in 2012, ** Approved by the FDA in 2015.

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
