# Peer review of "Recent Advances in Signaling Pathways Comprehension as Carcinogenesis Triggers in Basal Cell Carcinoma"

_jcm, 2020, doi:10.3390/jcm9093010_

Round 1
Reviewer 1 Report
This is a very exhaustive review paper on the pathogenic pathways of basal cell carcinoma. Although the argument is interesting lot of material has been already recently reviewed (Mol Carcinog. 2017 Dec; 56(12): 2543–2557., Sci Rep. 2020 May 14;10(1):8005., Target Oncol. 2019 Jun;14(3):253-267.)
The author fail to go indepth with non canonical pathways such as the Yippo-YAP , that is only superficially mentioned .
Infact the second part of the review is much more interesting (from subsection 4 on ) with can become the main part of the review itself
I suggest the author to focus more on their main topic interest ( inflammation and immune response to cancer es.Immunologic Characteristics of Nonmelanoma Skin Cancers: Implications for Immunotherapy. Am Soc Clin Oncol Educ Book. 2020 Mar;40:1-10.) or on antagonist of smoothened receptors resistance genes expression rather than repeat the information on the PATCH1 pathway that have been already covered throughly in the recent literature.
Reviewer 2 Report
The authors have tried to review the signalling pathways in BCC, but the result is an accumulation of facts from multiple papers. Some sections are very difficult to read and there are very important mistakes that I have found. I am not an expert in all pathways, and I am concerned that other errors may have scape me. I would recommend the senior authors of the paper to review it.
The authors are making repeatedly a very critical mistake, they confuse NMSC (non-melanoma skin cancer) with BCC. NMSC at least comprises two very different cancers: SCC and BCC. Moreover, any other type of skin cancer (sarcomas, mastocytomas, histocytosis, etc) could be included in the term. For BCC and SCC, the term keratinocyte carcinoma is better. In any case, papers related to NMSC may or may not be applicable to BCC. The authors should critically review their own paper.
Section 2 on page 2, line 44, should be classify by frequency. Rarely mutated genes will have small role in therapeutics.
Although resistance to treatment may be related to gene mutation, it is a bit unusual to discuss this here. The readers have not been introduced to the treatments yet.
Section 3 on page 3, line 96, requires a few drawings to help with the understanding of the molecules and pathways.
Figure 1. This is insufficient for the paper, and do not show many of the pathways discussed.
Page 1, line 41 "in two particular scenarios, immunosuppressed patients and patients with certain genodermatoses, such as Gorlin syndrome."
Page 2, lines 41 and onwards: genes and proteins should have different writing style. It is very difficult to know when the authors are talking about the protein or the gene.
Page 4, line 163, probably the authors want to say "MHC-I"
Page 4, line 182. NMSC? Approved for non-melanoma skin cancer? If this is the case, NMSC comprises many more cancers than just Keratinocyte Carcinomas (BCC and SCC). The authors should review this. I would be surprised that any of these HHi have been approved for anything else but BCC.
Page 5, Figure 1. It is very difficult to read.
Page 5, line 194. The Wnt pathway is not represented in Fig 1.
Page 6, line 234. What is the meaning of "a role in relapse by modulating the cell identity of the residual tumor mass"?
Page 6, line 236. The PI3K/AKT/mTOR pathway is not represented in Fig 1.
Page 7, line 286. EGFR is probably not very important if only 7% of HNSCC display EGFR mutations. Could the authors review this?
Page 7, line 306. That the treatment with cetuximab is useful in NMSC does not implies its role in BCC. The authors should review this.
Page 7, line 311. The activation of the MEK/ERK pathway is a classical mechanism for hyperproliferation (eg Melanoma and the SCC after the use of BRAF inhibitors). It is strange that the authors say that this is antiproliferative. This should be reviewed.
Page 7, line 312. NMSC?
Page 8, line 365. This pathway is not represented in Fig 1.
Round 2
Reviewer 1 Report
The authors have answered the point raised and have improved the paper .
The paper has still a considerable length but it is worth it .
Reviewer 2 Report
The changes have improved the paper.